# Acute and Chronic Effects of Supervised Flexibility Training in Older Adults: A Comparison of Two Different Conditioning Programs

**DOI:** 10.3390/ijerph192416974

**Published:** 2022-12-17

**Authors:** Stefano La Greca, Mariano Rapali, Giuliano Ciaprini, Luca Russo, Maria Giulia Vinciguerra, Riccardo Di Giminiani

**Affiliations:** 1Department of Biotechnological and Applied Clinical Sciences, University of L’Aquila, 67100 L’Aquila, Italy; 2Department of Human Sciences, Italian University Line—IUL, 50122 Florance, Italy

**Keywords:** flexibility, stretching, range of motion, older adults, sit-and-reach, acute effect, chronic effect

## Abstract

Flexibility training is a fundamental biological process that improves the quality of life of the elderly by improving the ranges of motion of joints, postural balance and locomotion, and thus reducing the risk of falling. Two different training programs were assessed acutely and after 12 weeks by means of the sit-and-reach test. Thirty-one healthy older adults were randomly divided into three groups: the Experiment I group (Exp) performed strength and static stretching exercises; the Experiment II group performed dynamic and static stretching exercises; and participants assigned to the control group maintained a sedentary lifestyle for the entire period of the study. Flexibility acutely increased in Exp I by the first (ΔT0 = 7.63 ± 1.26%; ES = 0.36; *p* = 0.002) and second testing sessions (ΔT1 = 3.74 ± 0.91%; ES = 0.20; *p* = 0.002). Similarly, it increased in Exp II significantly by the first (ΔT0 = 14.21 ± 3.42%; ES = 0.20; *p* = 0.011) and second testing sessions (ΔT1 = 9.63 ± 4.29%; ES = 0.13; *p* = 0.005). Flexibility significantly increased over the 12 weeks of training in Exp I (ΔT0 − T1 = 9.03 ± 3.14%; ES = 0.41; *p* = 0.020) and Exp II (ΔT0 − T1 = 22.96 ± 9.87%; ES = 0.35; *p* = 0.005). The acute and chronic differences between the two groups were not significant (*p* > 0.05). These results suggest the effectiveness of different exercise typologies in improving the flexibility of the posterior muscular chains in older adults. Therefore, the selection of a program to optimize training interventions could be based on the physical characteristics of the participants.

## 1. Introduction

Aging is a natural and biological process in which physical fitness and physiological characteristics change during the lifespan [1,2]. In this respect, flexibility plays an important role among the components of health in the elderly, counteracting disability and distress [3]. 

The lack of flexibility in the elderly can be both a cause and a consequence of postural imbalance, movement limitations and alterations in spatiotemporal parameters during gait (i.e., walking speed, stride length, frequency of gait and range of motion). These hamper daily activities and consequentially reduce quality of life [4,5]. Additionally, a range of motion (ROM) reduction can increase the risk of falling among middle-aged and elderly individuals [6]. 

Although a deficit in flexibility affects several kinetic chains in the elderly [7], the posterior kinetic chains of the body are most affected [1], probably because of the changes that occur in posture [3] and/or the time spent in a sedentary lifestyle [8]. There is a flexibility decrease in the upper and lower joints by approximately 6 degrees in those over 55 years of age, in both sexes, with each decade of life. In particular, decreases in hip flexion of 0.6 degrees per year in males and 0.7 degrees per year in females have been documented [9]. 

Tight hamstrings, low stretch tolerance, poor hip contracture, altered pelvic tilt and muscle–tendon stiffness [10] are the major factors responsible for the decrease in flexibility, and consequently for physiological changes, in aging; in particular, these modifications are induced by the neuromuscular system [11]—that is, endogenous alterations (i.e., changes in the sensitivity of peripheral nociceptors) [12] and the decreased collagen synthesis in different tissues (i.e., skin, ligaments, tendons and deep tissues) [1,13]. 

However, these alterations could be counteracted by performing stretching exercises [14,15]. Battaglia et al. [16] showed that elderly women who performed 8 weeks of training flexibility increased their spinal ROM; specifically, the sacral/hip joint ROM improved by 34 percent. Moreover, low back pain (LBP) in the elderly represents a highly disabling condition [17] and frequently is associated with limited flexibility of the lower limbs. It can be reduced by improving the flexibility and ROM of the trunk [18], which in turn decreases the risk of injury and determines an increase in quality of life (QoL) [19]. In addition, a recent study by Nishiwaki et al. [20] showed a relationship between changes in flexibility through a stretching training protocol and a reduction in arterial stiffness. 

In a review by Behm et al. [21], the different stretching typologies (i.e., static stretching (SS), dynamic stretching (DS), ballistic stretching (BS) and proprioceptive neuromuscular facilitation (PNF)) that influence the specific mechanisms responsible for acute increases in ROM have been reported. In several investigations, flexibility training resulted in an increased ROM, specifically in the hamstring muscle group, regardless of the stretching typology [22,23,24,25], whereas other studies underlined conflicting results when comparing SS, DS, BS and PNF [21,26]. However, SS is the most appropriate typology of stretching for sedentary, untrained and elderly subjects [24,27] to enhance joint ROM [21,28,29], as the muscle lengthening is obtained by slowly moving a joint to its maximal ROM, without increasing the reflex activity of the stretched muscle [30]. The SS technique acutely increases the muscle length by autogenous inhibition, eliciting the Golgi tendon organ’s activation [31], whereas the chronic changes are attributed largely to the reduced intrinsic stiffness of the muscle–tendon unit, followed by neural adaptations [30]. Interestingly, some investigations have confirmed that strength training, with appropriate loading, also improves the ROM in the elderly [32,33,34,35]. Barbosa et al. [32] suggested that 10 weeks of resistance training involving various exercises (seated chest press, seated row, seated shoulder press, seated curl—unilateral, seated triceps extension—unilateral, seated leg press, seated calf press and seated abdominal crunches) improved the sit-and-reach scores of elderly women without any additional stretching exercises. The mechanisms responsible for this improvement have not yet been clarified; probably, strength exercises cause a reduction in passive tension and stiffness of the tissues surrounding a joint [35]. In fact, these structures tend to limit the range of motion, as the cross-links increase during aging, making the connective tissue less compliant [1,36].

There are several studies and reviews that analyze and compare various stretching exercises or interventions to improve flexibility [22,24,25,26,29,37,38]. From a practical perspective, a comparison of flexibility conditioning programs, usually performed in fitness centers, combining static and dynamic stretching or static stretching and total body exercise (i.e., strength training in the form of a circuit training), is lacking in the literature. Unique combinations of different exercises should be considered the optimal solution in order to meet the needs of participants, as many people do not tolerate the intense sensory feedback (i.e., pain and discomfort) during the execution of static stretching exercises [39]. Therefore, we aimed to investigate the acute and chronic effects (12 weeks) on the flexibility of the posterior musculature induced by two different combined programs of conditioning in older adults. The assessment of the acute effect was used to optimize the training over time by dosing the single stimulus relative to the participant’s responsiveness. 

We hypothesized that, in older participants, the two combined conditioning programs (strength + static stretching vs. dynamic + static stretching) would improve the acute and chronic flexibility of the posterior muscular chains, which was assessed by the sit-and-reach test (SR) [40]. The static stretching, common to both programs, was analogous to the trunk bending required in the toe touch test, but the participants started from the standing position with the hands on the thighs to reduce the forces on the lumbar spine due to external torque. 

## 2. Materials and Methods

### 2.1. Experimental Design and Participants

A study with 3 parallel groups and repeated measures was performed. Thirty-one healthy older adults took part voluntarily in this study. The sample size estimation for the dependent variable (sit-and-reach score) was computed a priori by means of statistical software for power analysis (G*Power 3.1.9.4, Heinrich Heine-Dusseldorf University, Düsseldorf, Germany). The computation was performed in relation to the study design and using both parametric and nonparametric procedures, setting the effect size (ES) and using the protocol for a power analysis (test attributes, moderate ES [0.60–0.70], α = 0.05, power [1-β] = 0.95, sample size n = 30 participants). Participants were randomly divided into three groups and the allocation sequence was generated using the block randomization algorithm. This algorithm randomizes subjects to two or more groups and achieves balance across groups. The sequences were generated using statistical software (Pass 13-NCSS, LLC, Kaysville, UT, USA). Three groups were generated: the Experiment I group (Exp I); men: 7; women: 3; age: 63.7 (1.4) years; height: 1.69 (0.02) m; body mass: 79.3 (5.5) kg; body mass index: 27.6 (1.5) kg·m^−2^; the Experiment II group (Exp II); men: 4; women: 6; age: 67.9 (2.8) years; height 1.6 (0.03) m; body mass: 69.1 (2.5) kg; body mass index: 25.4 (0.8) kg·m^−2^; and the control group; men: 6; women: 5; age: 66.8 (1.7) years; height: 1.59 (0.03) m; body mass: 74 (4.2) kg; body mass index: 29.7 (1.1) kg·m^−2^. Each participant was informed about the procedures of the study and the relative risks. Everyone provided written informed consent. The participants of Exp I and Exp II executed supervised training programs similar to those conducted in fitness centers and reported in the practice guidelines [41]. The participants of the control group did not train and maintained a sedentary lifestyle for the entire period of the study (12 weeks). All the measurements were conducted in the Laboratory of Biomechanics of the Department of Applied Clinical Sciences and Biotechnology of the University of L’Aquila. The study was part of a project for the promotion of health in university staff (called “Ateneo in Movimento”). We excluded from the present study people who were affected by acute or chronic diseases, heart disease, a neurological disorder, a psychological disorder and low back pain. The intervention lasted 12 weeks, and flexibility measurements were carried out on all the participants. 

### 2.2. Testing Procedures 

To test the flexibility levels of the participants, we used the SR. The test provides a reliable measure of the degree of flexibility of the posterior muscular chains [42], and the measurements were carried out for all the participants in the following way: before and after the first session of training (T0); before and after the last session of training (T1 at the end of the 12-week training period). The values of flexibility measured before (Pre) the first training session at T0 and before (Pre) the last training session at T1 were used to assess the chronic effect (T0–T1), whereas the acute effects were assessed at T0 (pre–post the first training session) and at T1 (pre–post the last training session) (Figure 1). The assessment of the acute effect at T0 and T1 was used to determine the training dose at the beginning for 12 weeks of training. In each testing session, the mean value of 3 trials (separated by an interval of 20 s each) was retained for analysis. The measurements were performed at the same time of day to avoid circadian variations [43]. During the execution of the SR test, the participant sat on the floor with his/her legs stretched forward and the soles of the feet against a box; knees were locked and pressed flat to the floor; arms were stretched forward, above the box, with the palms facing downwards and the hands side by side. The participant flexed their trunk forward, moving the cursor along the measuring line as far as possible [44]. The maximum position was held for ~2 s, and then the participant returned to the starting position [44]. Ten minutes of warming up was performed by all the participants before the SR test and the training sessions. The warm-up was composed of 10 min of walking and 5 min of static and dynamic stretching exercises for the upper limbs, lower limbs and trunk.

### 2.3. Physical Training Interventions 

Over the training intervention (12 weeks), the participants of Exp I and Exp II performed 2–3 weekly sessions, and they were supervised by sports science graduate trainers. The exercise selections were based on the indications for older adults reported in the literature [13] and included exercises for the major muscle and joint groups.

Two different conditioning training sessions were conducted by the participants of Exp I and Exp II, but, in both groups, each training session lasted approximately 30 min: those in Exp I performed strength training exercises (bodyweight exercises or use of muscle-building machines with an overload that allowed participants to perform up to 10 repetitions) and static stretching (Table 1); those in Exp II performed dynamic and static stretching exercises. The movements during dynamic stretching were executed slowly to avoid reflex muscle activation [30] that could cause muscle–tendon injuries. 

For the static stretching programs, we selected a single exercise in which the posterior muscular chains were stretched: forward bending of the trunk from a standing position (Figure 2); this exercise, which involves trunk flexion and extension, increases the relaxation–stabilization of the muscles (i.e., latissimus dorsi and lumbar erector spinae) and reduces LBP [45]. Considering the age of the participants, particular attention was given to the stretching position to avoid pain or discomfort at the lower back. Therefore, we asked participants to place their hands on their thighs, in order to reduce the weight on the trunk and to minimize the action of compressive and shearing forces on the lumbar vertebrae segments (L4-L5), which is recognized as a risk factor for lower back pain [46,47,48]. The forward bending of the trunk was performed to the point of mild discomfort, following the ACSM indications for adults [23]: 2 sets of 10 repetitions, involving holding the position to the point of moderate discomfort in the hamstrings [25] for 15 s; 3 min of rest were taken between sets. The training workload was constant during the 12 weeks to compare the acute effects induced at baseline and after 12 weeks.

### 2.4. Statistical Analysis

Shapiro–Wilks’s W test revealed that the data were not normally distributed; therefore, the choice of a nonparametric statistical analysis allowed us to address the lack of Gaussian shape. 

The physical characteristics among the three groups were assessed by means of the Kruskal–Wallis test with multiple pairwise comparisons using the Steel–Dwass–Critchlow–Fligner procedure/two-tailed test. The acute and chronic effects of physical training (independent variable) on flexibility (dependent variable) were assessed in the two testing sessions by means of the Wilcoxon signed-ranked test/two-tailed test in each group. The two experimental groups were compared by using the Mann–Whitney test. The intrasession and inter-day reliability of the flexibility measurements were quantified using the intraclass correlation coefficient (ICC of single measures) of the log-transformed values [49]. In agreement with previous studies [50], values of ICC less than 0.50 were defined as “poor,” those from 0.50 to 0.69 were defined as “moderate,” those from 0.70 to 0.89 were defined as “high” and those greater than 0.90 were defined as “excellent”. The significance level was set to *p* < 0.05. All the analyses were executed using XLSTAT version 15 (Statistical Data Analysis Solution, Addinsoft, New York, USA 2022; https://www.xlstat.com). 

The statistical significance was set to *p* ≤ 0.05, and the meaningfulness of significant outcomes was estimated by calculating the ES of Cohen.

## 3. Results

The results of 29 older adults were analyzed in the two testing sessions before and after flexibility training as two participants in the control group dropped out of the investigations due to health problems. The participants did not report side effects or lower back pain, and none of the baseline measurements (age, body mass, stature, BMI and sit-and-reach) were significantly different among the three groups (*p* > 0.05). 

### 3.1. Reliability of Measurements

The intraclass correlation coefficients (ICCs, 95% confidence limit, lower confidence limit–upper confidence limit) of the measured variable between testing sessions T0 and T1 (chronic) were 0.96 (0.82–0.99), 0.93 (0.71–0.98) and 0.97 (0.87–0.99), for Exp I, Exp II and the control group, respectively. The ICC for T0 and T1 (acute) was 0.99 (0.96–1.00) for Exp I, 0.98 (0.93–0.99) for Exp II and 0.97 (0.91–0.99) for the control group.

### 3.2. Acute Effect

The acute effects are shown in Figure 3. In Exp I, flexibility acutely increased significantly over T0 (ΔT0 = 7.63 ± 1.26%; ES = 0.36; *p* = 0.002) and T1 (ΔT1 = 3.74 ± 0.91%; ES = 0.20; *p* = 0.002), whereas, between ΔT0 and ΔT1, it decreased (ΔT0 − T1 = −46.45 ± 11.16%; *p* = 0.008). Similarly, it increased significantly in Exp II during T0 (ΔT0 = 14.21 ± 3.42%; ES = 0.20; *p* = 0.011) and T1 (ΔT1 = 9.63 ± 4.29%; ES = 0.13; *p* = 0.005), but the difference between ΔT0 and ΔT1 (ΔT0 − T1 = −65.22 ± 64.21%) was not significant (*p* = 0.193). Conversely, the control group did not show significantly increased flexibility either in T0 (ΔT0 = 6.60 ± 3.77%; *p* = 0.089) or in T1 (ΔT1 = 12.20 ± 7.22%; *p* = 0.092). The difference between the two trial groups was not significant for T0 (6.78 ± 4.15%; *p* = 0.159) or T1 (5.89 ± 4.47%; *p* = 0.072).

### 3.3. Chronic Effect

The chronic effects are summarized in Figure 4 and Figure 5. Flexibility significantly increased in Exp I (ΔT0 − T1 = 9.03 ± 3.14%; ES = 0.41; *p* = 0.020) and Exp II (ΔT0 − T1 = 22.96 ± 9.87; ES = 0.35; *p* = 0.005) over the 12 weeks of training. The difference between the two experimental groups at the end of the intervention (T1) was not significant (17.67 ± 5.70%; *p* = 0.089). The control group did not show significant changes (ΔT0 − T1 = −3.23 ± 5.88%; *p* = 0.953). 

## 4. Discussion

The results of the present study confirmed our hypothesis that the two conditioning programs significantly induced flexibility improvements acutely and following 12 weeks of training in older adults. The acute and chronic changes between the two protocols were not significant. The magnitude of the training’s effects (effect size, ES) relative to the pre–post comparisons within each group was larger in Exp I than in Exp II, both acutely (0.20–0.36 vs. 0.13–0.20) and chronically (0.41 vs. 0.35). On the contrary, the relative change was the highest in Exp II, but there was greater variability in this group, whose participants performed dynamic and static stretching (Figure 3 and Figure 5). 

A possible explanation of the latter result could be found in the different categories of exercises (strength vs. dynamic stretching), as the static stretching was the same in the two groups. Strength exercises were executed with weight machines that had fixed ranges of motion, limited degrees of freedom for movement and high reproducibility [51]. Therefore, strength machines can be used easily by the elderly. Their standardized execution allows for more consistency than free movements, such as dynamic stretching exercises, which need to be controlled by the performer themselves, and experience is required to execute them correctly and effectively (i.e., amplitude and velocity of stretching) [52]. 

In the literature, researchers have studied the chronic effects induced by combinations of endurance and strength exercises on flexibility. They have placed less emphasis on flexibility exercises. Barbosa et al. [32] reported flexibility improvements (sit-and-reach test, 13 ± 9%) in elderly women following 10 weeks of strength training without stretching exercises. Additionally, Fatouros et al. [36] compared the effects induced by strength training, cardiovascular training and strength and cardiovascular training on several variables, including the sit-and-reach score, following 16 weeks of training (48 sessions). An increase in the sit-and-reach score was obtained with strength training and strength and cardiovascular training (~10–12%). However, stretching training alone was more effective than the combined training (strength and stretching) after 10 weeks (three sessions per week) in improving the hips’ ROM (14.5 vs. 5.5%) [33]. Higher improvements in hip ROM have been reported following 12 weeks (24 sessions, 60 min for each session) of pilates exercises (22%) [53]. Recently, Sobrinho et al. [54] showed that adding stretching exercises to multicomponent training for 16 weeks is very effective at generating additional benefits for other physical variables (i.e., strength, agility and aerobic fitness). 

Overall, these results underline that different typologies of exercises are effective at improving flexibility because the limiting factors are several and related to neural and muscular components [30]. Nevertheless, the choice of a specific training regimen should be made considering the physical characteristics of participants and their health conditions. 

In the literature, other investigations have reported the acute effects of stretching exercises on hip ROM [26] and sit-and-reach score [55], but studies are lacking that have analyzed both acute and chronic effects. In our study, we assessed the acute effect repeatedly, to check the effectiveness of the stimulus over time; this approach is fundamental to establishing the principle of progression of the training load and to ensuring specific adaptative responses in middle- and long-term flexibility training. In the present study, the acute effect of a single training session decreased significantly in the two experimental groups, indicating that the single “dose” could be increased for the elderly after 12 weeks of flexibility training. 

Additionally, the participants of both experimental groups performed a single static stretching exercise in which they bent forward the trunk in a standing position, similar to the movement during the toe touch test. The selected stretching exercise has three features. Firstly, this multi-joint exercise is an effective stimulus, as it involves the whole body and the stretching of several muscles (hamstring, lumbar, gluteus and triceps surae). Secondly, the participants, during the forward bending of the trunk, had their hands gripping their thighs, similarly to a “braced arm-to-thigh technique” [46,47,48]; the technique was adopted to oppose the external torque (given by the product of the trunk weight force and moment arm) to reduce the lumbar spine load (compression and shear forces) at the L4–L5 intervertebral discs [46,47,48]. We did not measure these forces because it was beyond the scope of this investigation. No adverse effects on the lower back were reported by the participants over the 12 weeks of training. Thirdly, the “braced arm-to-thigh technique” facilitated the pattern of lumbar flexion and pelvic rotation, which are the two main contributors to the forward bending of the trunk [56], while keeping the knees extended and hamstrings extended. 

The significant acute and chronic changes in flexibility resulted in the two conditioning programs, underlining that the exercises selected could induce similar neural and/or mechanical changes. Acute changes induced by the exercises performed could be due to the reduction in the reflex responses in the elderly [57]. The loading and holding phases of a stretch can increase the length of the muscle–tendon unit, thus modifying its viscoelastic properties [58]. The altered viscoelastic properties seem to affect the proprioceptive feedback and the motor-unit activation [59]. Additionally, the chronic changes in flexibility initially involve mechanical adaptations, followed by a reduction in neural input after 30 training sessions (an inverse sequence with respect to that observed during strength training) [30]. Additionally, the flexibility improvements have been attributed to increased stretch tolerance [39]; in other words, the intense sensation of discomfort when a muscle is stretched is reduced in the post-training period and could be associated with a reduction in the sensory feedback or an attenuated interpretation of the afferent signals [57]. 

### Limitations

In our study, the relative contributions of neural and muscular factors and the timing of these adaptations could not be discerned with regard to the different exercise typologies because no specific measurements were carried out. A point of concern in the present study regards the possible effects induced by resistance training exercises on the development of muscular strength of the upper and lower limbs, which we did not assess, as our aim was to determine their effect on flexibility. However, it could be useful to develop both strength and flexibility within a single session of training. 

## 5. Conclusions

The results showed a flexibility increase in both groups; therefore, the choice of a program that optimizes effectiveness and allows the prevention of any adverse effects should be based on the physiological and clinical characteristics of the elderly person [60]. The ACSM guidelines [61] suggest avoiding resistance exercises in subjects with recent heart disease (myocardial infarction or electrocardiographic changes, complete heart blockage, acute congestive heart failure, unstable angina or uncontrolled hypertension). In this population, stretching exercises (dynamic) could be used, which contribute to lowering both systolic and diastolic blood pressure [62]. Resistance training is also contraindicated in elderly persons with compromised bone health due to a comorbidity. For example, subjects with rheumatoid arthritis can safely benefit from non-weight-bearing exercises [63].

Future studies should consider using the results of the acute effects to dose the stimulus in longer-term investigations (from 3 to 6–12 months) to optimize the adaptation process. Moreover, muscular and neural variables should be measured to identify the mechanisms that limit flexibility in the elderly.

## Figures and Tables

**Figure 1 ijerph-19-16974-f001:**
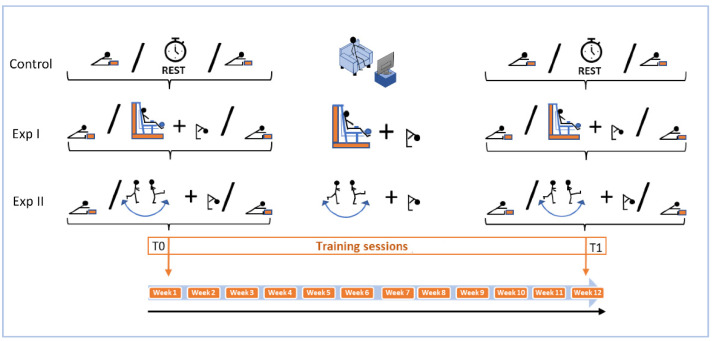
Timeline of exercises performed by the three groups during the follow-up interventions. Exp I: strength training combined with specific flexibility exercise; Exp II: dynamic flexibility training combined with specific flexibility exercise; control: maintained a sedentary lifestyle during the training period. In T0 and T1, we assessed the effect of a single session in the three groups. The control group had only a rest period to wash out the effect of the sit-and-reach test.

**Figure 2 ijerph-19-16974-f002:**
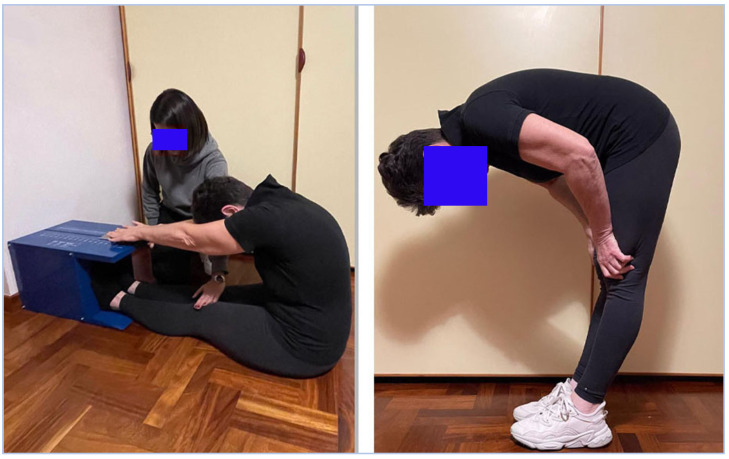
On the right: static stretching exercise performed by the participants of the two experimental groups; the trunk flexion is supported by the arms, which are gripped on the knees. On the left: sit-and-reach test to assess the flexibility.

**Figure 3 ijerph-19-16974-f003:**
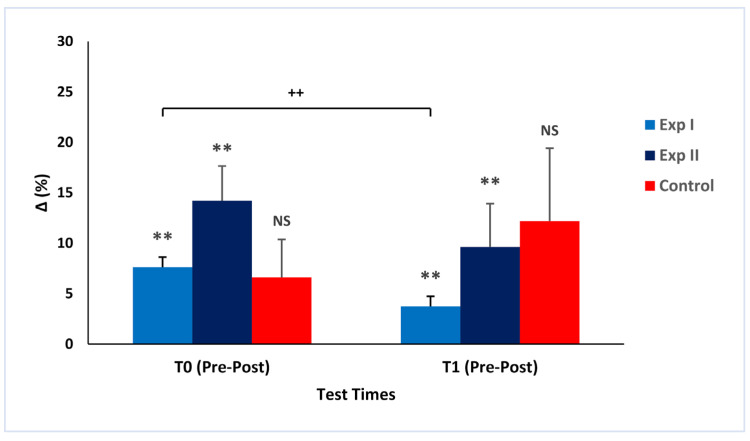
Mean values and standard deviations are shown. Acute relative changes in flexibility in the testing sessions T0 (baseline) and T1 (after 12 weeks). ** Significant changes within the group (*p* < 0.05); ^++^ significant changes between two groups (*p* < 0.01); NS—no significant changes (*p* > 0.05).

**Figure 4 ijerph-19-16974-f004:**
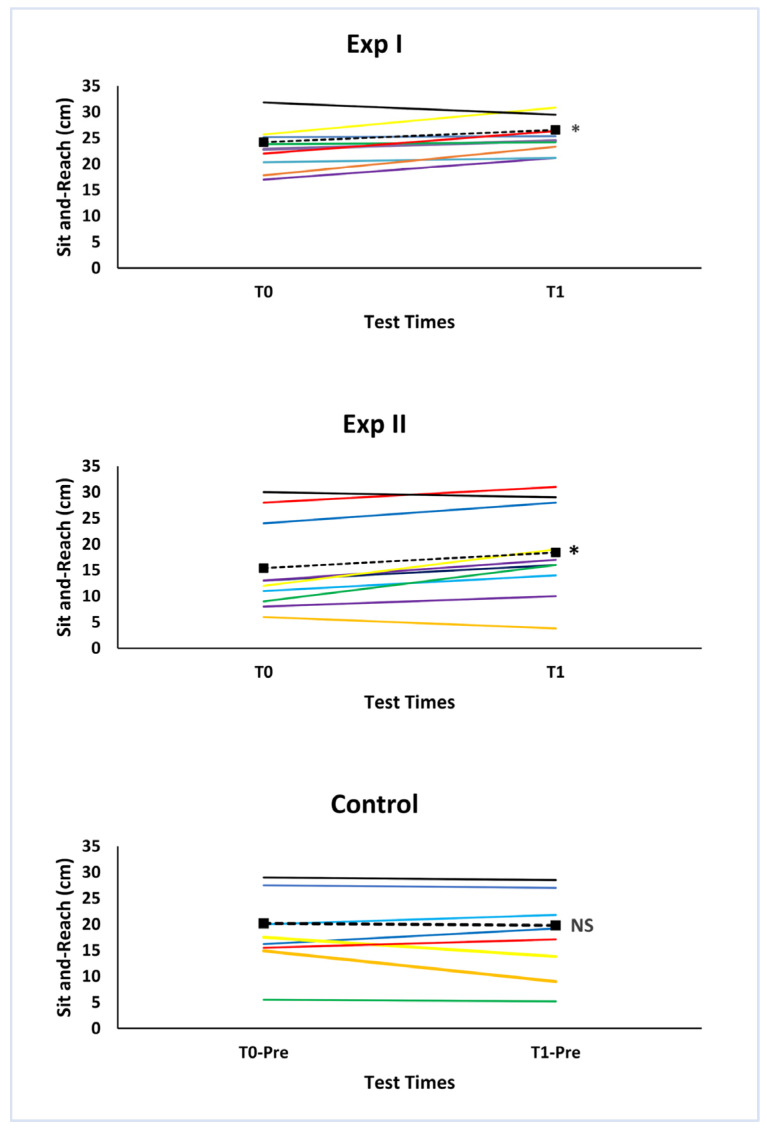
Chronic absolute changes over the 12 weeks (baseline—T0; T1—after the interventions) in the three groups. The solid lines of different color indicate individual changes, whereas the dashed lines indicate the mean value changes. * Significant changes (*p* < 0.05).

**Figure 5 ijerph-19-16974-f005:**
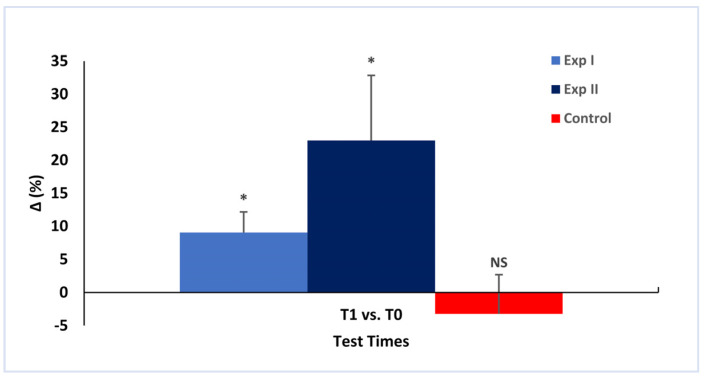
Mean values and standard deviations are shown. Chronic relative changes in the three groups. * Significant changes within the group (baseline: T0; T1: after 12 weeks of intervention; *p* < 0.05; NS: no significant).

**Table 1 ijerph-19-16974-t001:** Typologies of strength, dynamic and static stretching exercises in the two groups.

Exp I		Exp II	
**Strength Training Exercises**	**Sets × Reps ☆**	**Dynamic Training Exercises**	**Sets × Reps ☆**
(a) Pectoral machine/knee assisted push-up or wall push-up	3 × 10	(i) Hip external and internal rotator	4 × 10
(b) Seated lat. pull-down/bodyweight prone lat. pull-down	3 × 10	(j) Hip adductor and abductor	4 × 10
(c) Seated biceps curl	3 × 10	(k) Hip flexor and extensor	4 × 10
(d) Seated triceps extension unilateral/chair triceps dip	3 × 10	(l) Standing knee flexor	4 × 10
(e) Seated shoulder press	3 × 10	(m) Plantar flexor	4 × 10
(f) Bodyweight squat	3 × 10	(n) Trunk lateral flexion	4 × 10
(g) Bodyweight glute bridge	3 × 10	(o) Trunk rotator	4 × 10
(h) Seated calf press/bodyweight standing calf raise	3 × 10	(p) Shoulder flexion and extension	4 × 10
		(q) Shoulder girdle abduction and adduction	4 × 10
		(r) Shoulder external and internal rotation	4 × 10
**Static Stretching Exercise**	**Sets × Reps** **★** **/Time to SS (s)**	**Static Stretching Exercise**	**Sets × Reps** **★** **/Time to SS (s)**
(s) Bending the trunk forward with the hands on the knees	2 × 10/15	(s) Bending the trunk forward with the hands on the knees	2 × 10/15

Exp I = Experimental group I; Exp II = Experimental group II; trunk = a, b, n, o; upper limbs = c, d, e, p, q, r; lower limbs = f, g, h, i, j, k, l, m, s. SS = static stretching; reps = repetitions. ☆ Rest period between the sets was around 1 min. ★ Rest period between the sets was around 3 min.

## Data Availability

The dataset will be made available upon request.

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
