# Peer review of "Acute and Chronic Effects of Supervised Flexibility Training in Older Adults: A Comparison of Two Different Conditioning Programs"

_ijerph, 2022, doi:10.3390/ijerph192416974_

Round 1
Reviewer 1 Report
Thank you for the opportunity to review this paper. The paper is well-written and the contents are presented in great detail. I have a few minor comments that can be addressed to improve the paper.
First, the study and experimental trials involve human subjects, thus the study protocols are approved by the local ethical committee. I suggest the authors should also approve the study from the International Clinical Research Information Service [CRIS] with post trials registration. Second, the limitations of the study should be clearly mentioned in the conclusions part, some of the limitations are presented in the last paragraph of the discussion section, so I suggest moving them to the conclusions section. Please do provide future work as well in the conclusions section.
Lastly, some of the experiment sets are based on strength exercises/postures while the subjects are older adults. In many cases, older adults are unable to perform postures/exercises that require strength and endurance due to a lack of muscle strength or poor motor skills, or they suffer from fine motor impairment due to various chronic conditions. These points can also be mentioned in the method part. I suggest the authors read and provide/cite some justification from the paper (Interdisciplinary Co-Design Research Practice in the Rehabilitation of Elderly Individuals with Chronic Low Back Pain from a Senior Care Center in South Korea) in order to provide a better ground regarding the issues associated with older adults in such settings.
Author Response
Reviewer #1
Comments and Suggestions for Authors
Thank you for the opportunity to review this paper. The paper is well-written and the contents are presented in great detail. I have a few minor comments that can be addressed to improve the paper.
Author’s response
We thank the Editor and the Reviewers for reviewing our manuscript and providing thoughtful and constructive comments. As requested, please find below our responses to each of the comments. We have highlighted changes in the main document where appropriate in yellow.
We would kindly ask you to consider our answers and propose our paper to be accepted for publication.
Thank you very much in advance for your kind assistance.
Reviewer #1 Comment
First, the study and experimental trials involve human subjects, thus the study protocols are approved by the local ethical committee. I suggest the authors should also approve the study from the International Clinical Research Information Service [CRIS] with post trials registration.
Author’s response
Thank you very much for your suggestion. We have reported the approval of the ethical committee as requested by the editorial office of the journal, however if it will be possible to register our study (it is not a clinical study) after acceptance, we will provide it. We suspect that the time will not be short considering that our local ethical committee took 4 months to approve it (we need to publish our study as soon as possible like all researchers).
Reviewer #1 Comment
Second, the limitations of the study should be clearly mentioned in the conclusions part, some of the limitations are presented in the last paragraph of the discussion section, so I suggest moving them to the conclusions section. Please do provide future work as well in the conclusions section.
Author’s response
In the format of this journal the limitations are presented with a subsection at the end of discussion. Therefore, we have added a subsection, anyway we can move the limitations to the conclusion section.
We have also provided indications for future studies.
Reviewer #1 comment
Lastly, some of the experiment sets are based on strength exercises/postures while the subjects are older adults. In many cases, older adults are unable to perform postures/exercises that require strength and endurance due to a lack of muscle strength or poor motor skills, or they suffer from fine motor impairment due to various chronic conditions. These points can also be mentioned in the method part. I suggest the authors read and provide/cite some justification from the paper (Interdisciplinary Co-Design Research Practice in the Rehabilitation of Elderly Individuals with Chronic Low Back Pain from a Senior Care Center in South Korea) in order to provide a better ground regarding the issues associated with older adults in such settings.
Author’s response
We have reported this point in the method section with an appropriate reference.
Reviewer 2 Report
Congratulations for the paper presented, we believe it is a good work, but we still need to clarify some methodological aspects to be able to approve its publication.
These aspects would be the following:
Line 107: "Thirty-one healthy older adults participated in this study". Why? Have you calculated the statistical power of this sample?
Line 108: "Participants were randomly divided into three groups". Can you explain what method of randomisation you used?
Line 193: The use of the Wilcoxon signed test is only justified when there is a fairly severe deviation from a normal distribution, is this the case?
Methodology: No cases were lost during the 12 weeks of the study?
Line 208: In line 107 it says 31 samples, but in line 208 it says "Results from 29 older adults were analysed", where are the 2 missing cases?
Author Response
Reviewer #2
Comments and Suggestions for Authors
Congratulations for the paper presented, we believe it is a good work, but we still need to clarify some methodological aspects to be able to approve its publication.
Author’s response
We thank the Editor and the Reviewers for reviewing our manuscript and providing thoughtful and constructive comments. As requested, please find below our responses to each of the comments. We have highlighted changes in the main document where appropriate in yellow.
We would kindly ask you to consider our answers and propose our paper to be accepted for publication.
Thank you very much in advance for your kind assistance.
Reviewer #2 comment
Line 107: "Thirty-one healthy older adults participated in this study". Why? Have you calculated the statistical power of this sample?
Author’s response
“The sample size estimation for the dependent variable (sit and-reach score) was computed a priori by means of a statistical software for power analysis (G*Power 3.1.9, Heinrich Heine-Dusseldorf University). The computation was performed considering the study design and using both parametric and non parametric procedures, setting the effect size (ES) and using the protocol for a power analysis (test attributes, ES = 0.60-0.70, α = 0.05, and power [1-β] = 0.95).
Reviewer #2 comment
Line 108: "Participants were randomly divided into three groups". Can you explain what method of randomisation you used?
Author’s response
The random allocation sequence was generated using the block randomization algorithm. This algorithm randomizes subjects to two or more groups and achieves balance across groups. The sequences were generated using statistical software (Pass 13-NCSS, LLC Kaysville, Utah 84037, USA).
Reviewer #2 comment
Line 193: The use of the Wilcoxon signed test is only justified when there is a fairly severe deviation from a normal distribution, is this the case?
Author’s response
Shapiro-Wilks’s W test revealed that the data were not normally distributed, therefore the choice of a non parametric statistical analysis allowed us to address the lack of gaussian shape. We have included in the statistical section.
Reviewer #2 comment
Methodology: No cases were lost during the 12 weeks of the study?
Author’s response
We forgot to report that two subjects of the control group that dropped out.
Reviewer #2 comment
Line 208: In line 107 it says 31 samples, but in line 208 it says "Results from 29 older adults were analysed", where are the 2 missing cases?
Author’s response
Two participants of the control group dropped out. Now, we have corrected in the text.
Reviewer 3 Report
Thank the authors for your efforts. This is an interesting research paper, which is well-structured and provides sufficient background data and statistical analysis. The results may contribute to both practice and theory. Here, however, are some suggestions as follows:
Please re-check the formatting and layout (e.g., Table 1). Please also strengthen the quality and interpretation of the picture (e.g., Fig. 2). May consider add more pics in Fig 2.
Author Response
Reviewer#3 general comment
Thank the authors for your efforts. This is an interesting research paper, which is well-structured and provides sufficient background data and statistical analysis. The results may contribute to both practice and theory. Here, however, are some suggestions as follows:
Author’s response
We thank the Editor and the Reviewers for reviewing our manuscript and providing thoughtful and constructive comments. As requested, please find below our responses to each of the comments. We have highlighted changes in the main document where appropriate in yellow.
We would kindly ask you to consider our answers and propose our paper to be accepted for publication.
Thank you very much in advance for your kind assistance.
Reviewer# 3 comment
Please re-check the formatting and layout (e.g., Table 1).
Author’s response
We have reformatted the Table 1.
Reviewer #3 comment
Please also strengthen the quality and interpretation of the picture (e.g., Fig. 2). May consider add more pics in Fig 2.
Author’s response
We have changed the figure 2 and added another one picture.